# Long-Term Culture of Canine Ocular Cells That Maintain Canine Papillomaviruses

**DOI:** 10.3390/v14122675

**Published:** 2022-11-29

**Authors:** Dan Zhou, Aibing Wang, Sarah Maxwell, Richard Schlegel, Hang Yuan

**Affiliations:** 1Department of Pathology, Georgetown University Medical School, Washington, DC 20057, USA; 2Oregon Veterinary Referral Associates, Springfield, OR 97477, USA

**Keywords:** canine, papillomavirus, ocular, conditional reprogramming (CR), CPV

## Abstract

Canine ocular papillomas occur on the haired skin of eyelids, conjunctival epithelium, and rarely on the cornea. Using PCR typing assays with canine papillomavirus type-specific primer sets, our study confirmed that the papillomas contained canine papillomavirus type 1. The positive result from a rolling circle amplification assay indicated the CPV1 viral genome in the cells is a circular episomal form. We also successfully established the first canine corneal cell line using the conditional reprogramming method. The cells exhibited an epithelial cell morphology, grew rapidly in vitro, and could be maintained long term. For the continued growth of the canine corneal cells, feeder cells played a more important role than Rho-kinase inhibitor Y-27632. More importantly, the viral CPV1 genome was maintained in the canine corneal cells during the long-term expansion. Unlimited supplies of these cells provide as a model for the study CPV in dog cells, and a platform for drug screening for effective therapies against canine papillomavirus infection in the future.

## 1. Introduction

In dogs, transmissible warts were first noted in 1898, and a viral etiology was established in 1959 [1]. The first canine papillomavirus (CPV), Canine Papillomavirus type 1 (CPV1, also refereed earlier as Canine Oral Papillomavirus (COPV)), was fully sequenced in 1994 [2]. In the years since these initial studies of CPVs, additional CPV types have been identified and PVs have been associated with an expanded range of canine diseases [3,4,5,6,7,8]. According to the PapillomaVirus Episteme (PaVE, http://pave.niaid.nih.gov/#home, accessed on 10 November 2020), currently there are total 23 types of CPVs, which have been sequenced and annotated.

The CPVs in dog infections were found mainly CPV1 and CPV2. CPV1 dominated in oral infections, while CPV2 dominated in cutaneous papillomas. Co-infections with CPV1 and CPV2 accounted for about 20% of all detected infections [9]. Canine ocular papillomatosis cases are rarely described. Ocular papillomas occur on the haired skin of eyelids, conjunctival epithelium, and rarely on the cornea. All are benign and generally cured with surgical excision.

Some studies have been associated these ocular lesions with papillomavirus infection [10,11,12,13,14]. Some other studies were not able to detect canine papillomaviruses in conjunctival squamous papillomas [15].

Established in 2012, the conditional cell reprogramming (CR) method is a simple cell co-culture technology with a Rho kinase inhibitor, Y-27632, in combination with fibroblast feeder cells [16]. The technology enables us to rapidly expand both normal and malignant epithelial cells from diverse anatomic sites and different mammalian species without the requirement of transfection with exogenous viral or cellular genes. Establishment of CR cells from both normal and tumor tissue is highly efficient. The robust nature of the technique is exemplified by the ability to produce 2 × 10^6^ cells in five days from a core biopsy of tumor tissue [17]. Additionally, these epithelial cells can be propagated indefinitely in vitro, yet maintain the capability to become fully differentiated when transferred into conditions that mimic their biological environment [18]. We have successfully established human papillomavirus (HPV)-positive cell models from HPV induced diseases. These models allow us to perform viral and cellular genetic analysis, and biological assays, including chemosensitivity testing as a functional diagnostics and therapeutic tool for precision medicine [19,20,21].

In this study, we successfully isolated and propagated the first canine ocular cell line from canine corneal papillomas using the conditional reprogramming method. These canine cells were rapidly reprogrammed and acquired long term expansion ability. Canine papillomavirus type 1 was detected in the papillomas. The presence of the CPV1 was maintained during the passages of these conditional reprogrammed canine cells.

## 2. Materials and Methods

### 2.1. Canine Cell Isolation and Propagation

Canine ocular papilloma was minced and digested with collagenase (StemCell, Canada) and trypsin. Dispersed single cells were culture under three different conditions: (1) with feeder cells (irradiated mouse fibroblast 3T3 cells, kindly gifted by Dr. Howard Green’s Lab [22]) in F media with 10 uM Y-27632 Rho-kinase inhibitor (Enzo Life Sciences, Farmingdale, NY, USA); (2) with feeder cells in F medium without Y-27632; or (3) in F media with 10 uM Y-27632. All cells were maintained in an incubator at 37 °C with 5% CO_2_. Cell cultures were splitted when a confluence approximately 80% was reached. The cell growth curve was plotted as accumulated population doublings (PDs) versus time (days).

### 2.2. DNA Isolation and Polymerase Chain Reaction (PCR) Amplification for CPV Typing

Total DNA was isolated from the canine biopsy specimens or cells by using DNeasy Blood & Tissue Kit (Qiagen, Germantown, MD, USA) according to the manufacturer’s instructions. CPV type-specific primers were designed for CPV type 1 to type 10 according to DNA sequences for them obtained from Genebank (http://www.ncbi.nlm.nih.gov/genebank/, accessed on 10 November 2020) (Table 1). PCR was performed on a DNA thermal cycler under the following conditions: denaturation at 95 °C for 10 min, 25 cycles of 30 s at 95 °C, 30 s at 60 °C and 30 s at 68 °C, with final extension at 68 °C for 10 min. PCR products were analyzed by electrophoresis on a 1% agarose gel stained with ethidium bromide, and band size was estimated according to a 1 kb MassRuler DNA ladder Mix (Thermo Scientific MassRuler DNA Ladder Mix). CPV types were determined based on the amplification products. Type-specific PCR products were cloned into pGEM-T easy vector (Promega, Madison, MI, USA) and sequenced for DNA fragment identity.

### 2.3. Rolling Circle Amplification Coupled with RFLP

Viral DNAs papillomaviruses were amplified by rolling circle amplification (RCA) (Rector et al., 2004) using a TempliPhi Amplification kit (General Electrics Biosciences). 20 ng of extracted viral DNA was used as template for the RCA which was performed with slight modifications to the manufacturer’s instructions by adding extra 0.5 µL of 10 mM deoxynucleotide triphosphates (dNTPs) in each 10 µL reaction system. The reactions were carried out under the conditions: heat denaturation for 5 min and incubation at 30 °C for 20 h in the presence of phi29 polymerase. The RCA products were then digested with BamH I, EcoR I, EcoR V, Hind III or Nhe I (Thermo Scientific) at 37 degree for 15 min. The digested DNA fragments from each reaction were separated on a 1% agarose gel. Each type of CPV has its own digestion pattern, restriction fragment length polymorphism (RFLP).

### 2.4. Realtime PCR Viral Copy Number

Quantitative real-time PCR was performed by using TaqMan reagent (Bio-Rad) on a Bio-Rad iCycler MyiQ, primers and probes (Table 2) used for the quantitation of E6 gene from CPV1. 5 µL of template DNA pBR322.CPV1 at a concentration of 0.8 pg/µL equating to 3 × 10^5^ copies, and five 10-fold serial dilutions, were used to establish calibration curves. Real-time PCR reactions were done in triplicate and the results were averaged for each case. The levels of CPV DNA were analyzed using IQ5 software according to the manufacturer’s (Bio-Rad’s) guidelines. Canine CYTB gene (Table 2) was used to determine the canine cell equivalents of each sample under qPCR analysis. CPV DNA load values were reported as viral copies per canine cell equivalents (copy/cell).

## 3. Results

### 3.1. Diagnosis of Corneal Lesion as Viral Papilloma

A ten-year-old spayed female Boxer presented with a grey mass on temporal bulbar conjunctiva OD. The lesion was removed by surgery under topical anesthesia. The biopsy was categorized as squamous papilloma. The dog returned 37 days later with a large grey mass on cornea and several conjunctival growths. Corneal lesion (0.5 cm diameter) was surgery removed (Figure 1A). The tissue was cut and separated into smaller pieces for histology, isolation of DNA for virus typing and generation of a canine ocular papilloma cell line using the conditional reprogramming method. Histologic examination of the lesion revealed papillomatosis with focal koilocytotic atypia, a finding consistent with viral papillomas (Figure 1B). The surface epithelium had a sharply delineated, unencapsulated, exophytic area of proliferation. The surface had a papillary appearance with a large amount of compact keratin. The underlining epithelium was mildly disorganized with clumped keratohyalin granules. There were many small aggregates of keratinocytes with clear cytoplasm, peripheralized chromatin and amphophilic intranuclear inclusion material (koilocytes). There was no evidence of squamous extension beyond the basement membrane.

### 3.2. Detection of Caine Papillomavirus in Canine Ocular Papilloma

Using CPV type–specific primers (Table 1), we identified Canine Papillomavurus type 1 (CPV1, also called Canine Oral Papillomavirus originally, COPV) in the lesion (Figure 2A). Sequencing of the PCR product showed that the product matched CPV1 genome. To further evaluate the CPV infection, we used Rolling Circle Amplification to amplify episomal viral DNA. The RCA products were then digested with BamH I, EcoR I, EcoR V, Hind III or Nhe I. The digested DNA fragments from each reaction were separated on a 1% agarose gel. The RFLP profiles of DNA sequences of CPV1 (GenBank accession number, NC_001619) was analyzed using Bechling (https://benchling.com, accessed on 10 November 2020). The pattern of restriction-enzyme digestion (BamHI, EcoRI, EcoRV, HindIII, or NheI) of the amplified product did match that anticipated for CPV1 Figure 2B.

### 3.3. Establishment of Canine Ocular Papilloma Cell Line with Conditional Reprogramming Method

Approximately 0.1 cm diameter biopsy was used for the generation of a canine cell line. The biopsy was processed by incubating them in a mixture of dispase and collagenase to enable physical separation of epithelium and dermal tissue. Later, the epithelium was dispersed into single cells by digestion with collagenase/trypsin. The resulting cells were plated on a bed of irradiated Swiss 3 T3 J2 cells (feeder cells) and F medium supplemented with 10 μM Y-27632 (Rho-kinase inhibitor) as previously described [17]. Cobble-stone shaped keratinocyte colonies (Figure 3, black circles) were readily visible after 3 days, and cultures reached confluence in 5 days. After the first plating, the keratinocytes were seeded at 4.0 × 10^3^ cells/cm^2^ and passaged every 3–5 days.

### 3.4. Feeder Cells Required for the Rapid Growth of Canine Ocular Papilloma Cell Lines

To define the optimal in vitro growth conditions to cultivate and propagate canine ocular cells, we evaluated the cell growth in three conditions: 1. F medium + feeders + Y-27632 (CR condition); 2. F medium + feeders; 3. F medium + Y-27632. All cells were incubated in 37 °C degree with 5% CO_2_ incubator. 1 × 10^5^ cells were seeded in a 75 cm^2^ flask per passage and split when a confluence approximately 80% was reached. Total cell number was counted during each passage. The full CR conditions, feeders + F medium + Y-27632, supported canine ocular cells growth beyond 25 population doublings (PDs) with an average growth rate of 1.24 days/doubling (Figure 4A) on day 31. The condition without Y-27632 Rho-kinase inhibitor, F medium + feeders, also support cell proliferation beyond 22 PDs on day 30 with an average growth rate of 1.36 days/doubling. Condition without feeders, F medium + Y-27632, supported cell proliferation in a less degree at 10 PDs on day 31 with a much slower rate of 3.10 days/doubling. There were no significant morphological differences among those three growth conditions (Figure 4B). Thus, the canine ocular cells can proliferate long term under the condition of F medium + feeders with or without Rho-kinase inhibitor.

### 3.5. Canine Ocular Papilloma Cell Lines Maintained with Canine Papillomaviruses

Human papillomavirus viral load quantification assays for cervical cancer cases have been established [23,24]. Studies of HPV load address the potential utility of predicting the progression or severity of the human disease [25,26]. In this study, we designed a quantitative realtime PCR assay for CPV1. Primers and Taqman probes (described in Table 2) were designed to target CPV E6 regions. A standard curve was employed using 10-fold dilutions, from 300,000 to 30 copies, of specific recombinant plasmid to calculate the viral DNA load. Canine CRPS gene (Table 2) was used to determine the canine cell equivalents of each sample under qPCR analysis. The canine ocular cells were all positive of CPV1 through the passages in all three growth conditions. For the cells growing in the full CR conditions, feeders + F medium + Y-27632, there were on average 0.6 copies of CPV per cell, and there were no significant changes in the level of viral copies between days 10, 20 and 30 (Figure 5). We observed similar levels of viral copy for the feeder + F medium growth condition. However, the viral copy was on average 0.3 CPV/cell, and it was about half of the other two growth condition.

## 4. Discussion

Canine ocular papillomatosis cases are rarely described. Ocular papillomas occur on the haired skin of eyelids, conjunctival epithelium, and rarely on the cornea. Papillomaviruses, CPV1, had been detected in cases associated these ocular lesions [10,11,12,13,14]. However, especially in conjunctival squamous papillomas, there was no detection of any canine papillomaviruses [15]. Our study is the first confirmed case for the presence of canine papillomaviruses in canine corneal papillomatosis. The PCR typing results using type-specific primer sets proved the infection was caused by canine papillomavirus type 1. Moreover, the positive result from RCA assay indicated the CPV1 viral genome remained in the cells as a circular episomal form.

In this study we successfully established the first canine corneal cell line using conditional reprogramming method. The cells in epithelium cell morphology grow rapidly in vitro, and can be maintained for long term in the CR condition. In the CR condition, both feeders and Y-2763 play critical roles in cell growth [16]. Comparing all three growth conditions we used, cells grew more rapidly in the conditions with the feeders. Cells in feeder-free condition, F Medium + Y-27632, grew only half the speed (Figure 4A). Therefore, for canine corneal cell, feeder cells play a more important role in supporting of the cell growth. It has been suggested that various growth factors that released from feeder cells support the robust growth of the CR cells [17,18]. On the other side, Rho-kinase inhibitor can help cells against cell senescence in CR condition. However, it appeared that the Rho-kinase inhibitor, Y-2763, was less important for the growth of canine corneal cell, since there was no difference in growth with the feeder conditions with or without Rho-kinase inhibitor.

Naturally papillomavirus-infected cell lines from anatomically distinct lesions are urgently needed and important for virus research and antiviral drug therapy development. Besides a few HPV16 and HPV18-positive cervical cancer cell lines (HeLa, SiHa, etc.), there are only two naturally papillomavirus-positive human cell lines (W12 and CIN612) from CIN lesions [27,28]. In the current study, we were able to establish the first canine CPV-positive cell line from CPV-infected lesions. More importantly, the level of CPV genome is maintained during the long-term passages of the cells. Unlimited supplies of these cells provide us as a model for the study CPV in dog cells, and a platform for drug screening against this canine papillomavirus infection in the future.

## Figures and Tables

**Figure 1 viruses-14-02675-f001:**
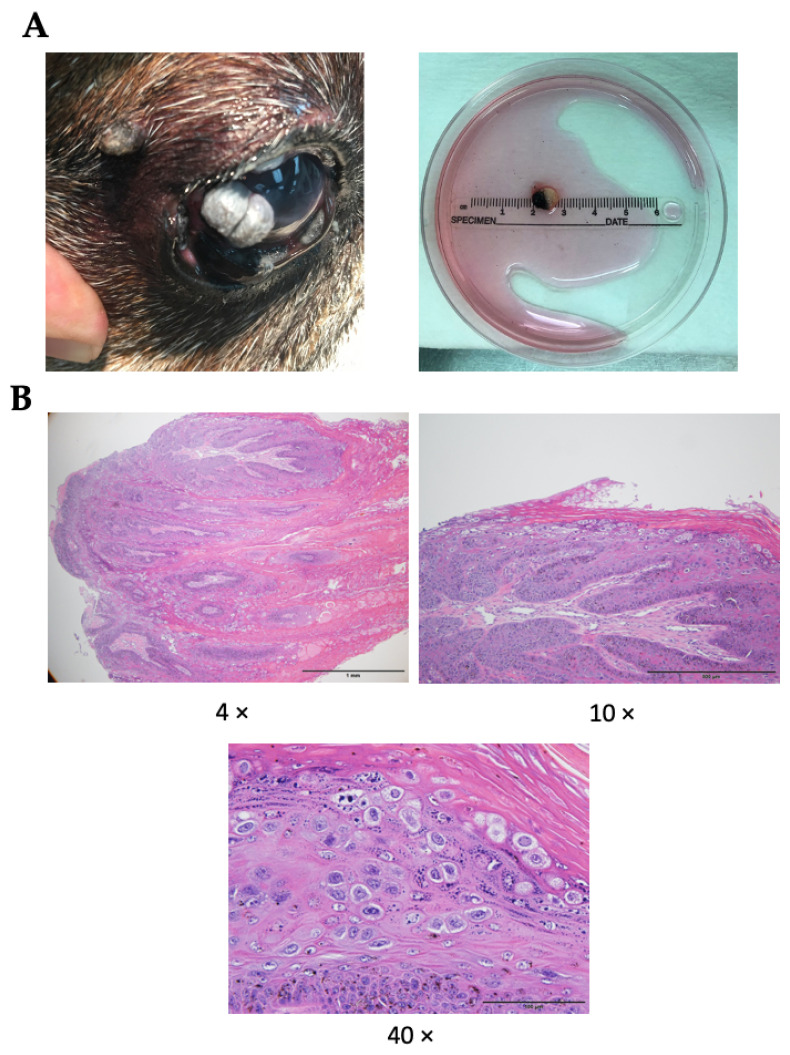
Canine ocular lesions. (**A**). A large grey mass on cornea and several conjunctival growths. Corneal lesion (0.5 cm diameter) was surgery removed. The tissue was cut and separated into smaller pieces for histology, isolation of DNA for virus typing and generation of a canine ocular papilloma cell line using the conditional reprogramming method. (**B**). Histologic examination of the lesion revealed papillomatosis with focal koilocytotic atypia, a finding consistent with viral papillomas.

**Figure 2 viruses-14-02675-f002:**
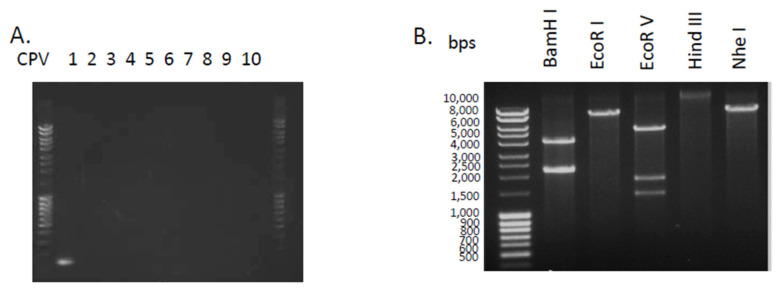
Detection of canine papillomavirus type 1 using DNA amplification. (**A**). Detection of CPV using type-specific CPV primer sets. DNA was isolated from biosy. PCR reactions were performed with type-specific primers: lane 1: CPV1; lane 2: CPV2; lane 3: CPV3; lane 4: CPV4; lane 5: CPV5; lane 6: CPV6; lane 7: CPV7; lane 8: CPV8; lane 9: CPV9 and lane 10: CPV10. (**B**). Canine papillomavirus typing using Rolling Cycle Amplification coupled with restriction fragment length polymorphism. The viral genomes were amplified by RCA, followed by the digestion indicated restriction-enzymes. The unique digestion patterns matched with CPV1.

**Figure 3 viruses-14-02675-f003:**
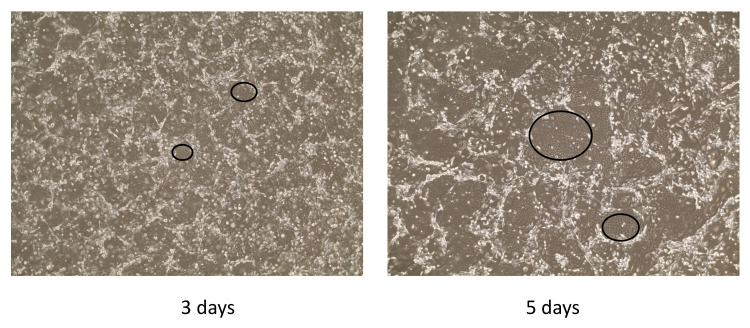
Isolation and establishment of canine corneal cells from a canine corneal papilloma. The morphology of the cells (co-culture with feeder cells). The images were taken with the phase contrast microscope on day 3 and 5. The circled areas were the examples of canine epithelial cells.

**Figure 4 viruses-14-02675-f004:**
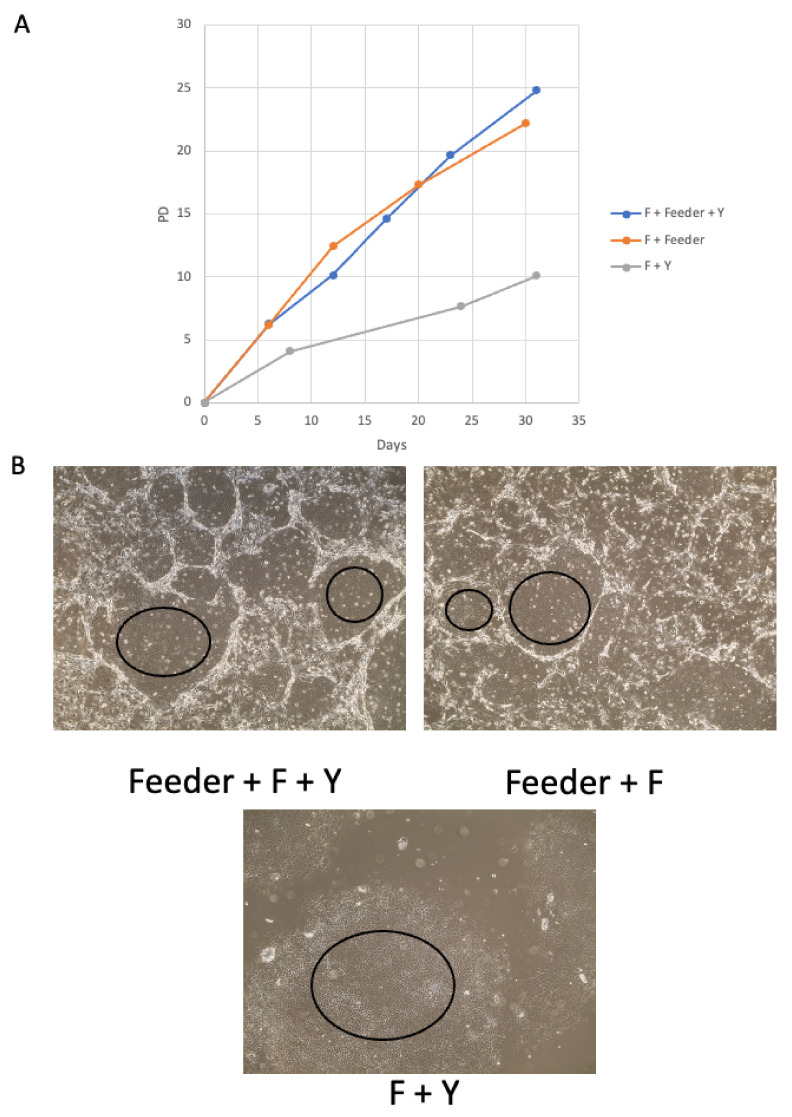
The long-term expansion of the canine corneal cells. The cell growth was evaluated in three conditions: 1. F medium + feeders + Y-27632 (CR condition); 2. F medium + feeders; 3. F medium + Y-27632. (**A**). The growth curve of canine corneal cells. The numbers of the cells were counted for each passage. Accumulated population doublings (PDs) were plotted with growth days. (**B**). The morphology of the cells under three different conditions on day 30. The circled areas indicated the examples of canine epithelial cells.

**Figure 5 viruses-14-02675-f005:**
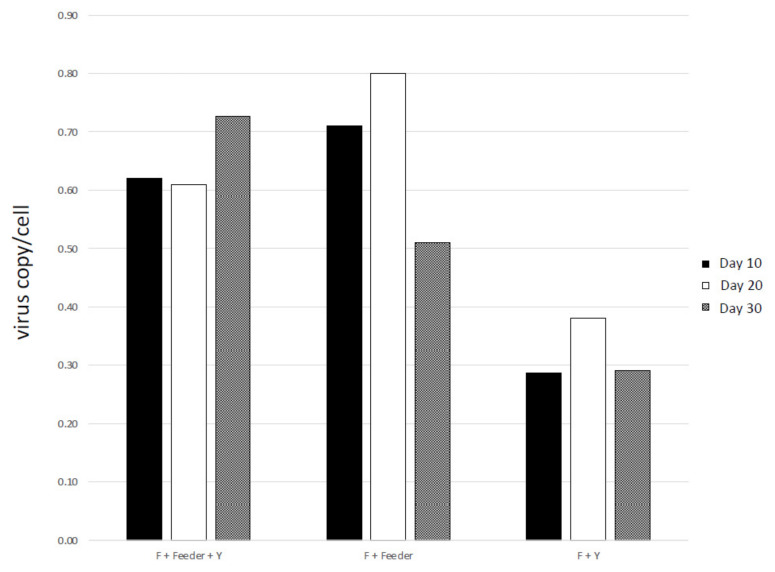
The canine corneal cells retained canine papillomavirus type 1 during the growth in all three different conditions. DNA was extracted from the cells. A recombinant plasmid pBR322.CPV1 was used to generate standard curve. Real-time PCR reactions were done in triplicate and the results were averaged for each case. Canine CYTB gene was used to determine the canine cell equivalents of each sample under qPCR analysis. CPV DNA load values were reported as viral copies per canine cell equivalents (copy/cell).

**Table 1 viruses-14-02675-t001:** Primer sequences for canine papillomavirus typing sets.

CPV typing 1 F	ACA AAA ATC TTG CTG GAC AT
CPV typing 1 R	TCC TGT GCA CCA TCT AGT TA
CPV typing 2 F	AAA GGT TGC AAC CAC AGA TGA CC
CPV typing 2 R	CGA GGG ACC TGA TTG TCA CA
CPV typing 3 F	AGT ATC AGC AGC AGA TGG CC
CPV typing 3 R	CTC CAT TGC AAG GCC TAG CA
CPV typing 4 F	TAC TAA GGA GGG TAA GGA TGC
CPV typing 4 R	CTT GAC AAC CTT ACC GCA GG
CPV typing 5 F	AGT ATC AGG ACA AGG CCC AG
CPV typing 5 R	AAC TGC ACC ATT ACA GGC GG
CPV typing 6 F	ATA AAA ACC CAA CTG ACC AT
CPV typing 6 R	ACA CAC CAA CGA GCT TGT AT
CPV typing 7 F	ACA CAT CTA GCA GCA AAG AT
CPV typing 7 R	TTG AGG CTC CTG ATC CTT AC
CPV typing 8 F	ATT ATG ACC CTG CAC AAG GT
CPV typing 8 R	ACA GAG ACA TTG GTG CAG GG
CPV typing 9 F	AGG TAT CAG CAG GGT GCG CA
CPV typing 9R	TGA CTC TTG CAG GGG GAG GC
CPV typing 10 F	GGG GGT GGA AAA GCA GCC TA
CPV typing 10 R	CTT GGT TCC ATT GCA TGC CC

**Table 2 viruses-14-02675-t002:** Real time assays for canine papillomavirus copy number.

	Forward Primer	Reverse Primer	Probe
CPV1	5′-TTGGAAGCCACAGACACTTG-3′	5′-TGTCCGCCTCACTCAGAATA-3′	5′-CAACGTAACAAAGACCTTGCAGCAGC-3′
CYTB	5′- CCTTACTAGGAGTATGCTTG -3′	5′- TGGGTGACTGATGAAAAAG -3′	5′- AAGTGGACTTGCACTATACATCGGACACAGCCA -3′

## Data Availability

Data sharing not applicable to this article as no datasets were generated or analyzed during the current study.

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
