# Peer review of "Long-Term Culture of Canine Ocular Cells That Maintain Canine Papillomaviruses"

_viruses, 2022, doi:10.3390/v14122675_

Round 1

Reviewer 1 Report

Dear Editor of the Journal Viruses

 General comments:

Regarding the manuscript viruses-2030111, entitled “Long-term Culture of Canine Ocular Cells That Maintain Ca-2 nine Papillomaviruses” I find the manuscript interesting and within the scope of Viruses published manuscripts. The authors developed a highly valuable tool to study Canine Papillomavirus and I would consider the manuscript for publication without relevant corrections.

 There are same minor spelling mistakes, such as:

Line 75 and 88- Different spelling of Celsius degree (line 75: oC; line 88 oC-superscript)

Line 87: deoxynocleotide instead of deoxynucleotide

Line 163: 1x105 cells. The 5 is not in superscript

Line 210: Consider the removing of the “one” (Our study is the first one confirmed case…”

My major remark concerns the RFLP pattern to be obtained with the described restriction enzymes. This pattern is not described nor referred by other authors. I believe that its inclusion would be important for a full understanding of figure 2B and the following results.

Author Response

Thank you for the positive feedback.  We also greatly appreciate your very helpful suggestions.

  • We have put more details in the RCA/RFLP experiment. Hopefully the revision is easier to interpreted by the readers;
  • Line 75 and 88- Different spelling of Celsius degree: all have been changed to oC;
  • Line 87: deoxynocleotide was changed to deoxynucleotide;
  • Line 163: 1x105 cells. Changed to 1x105 cells;
  • Line 210: Removed “one”

Reviewer 2 Report

Authors have done characterization of CPV and established the Canine Ocular Papilloma Cell Line with Conditional Reprogramming Method. Authors have carried out important studies to establish cell lines by using new techniques for the first time and this tool can be useful in future as well. It adds the new development in the cell line specifically canine ocular cell line. Authors have sufficiently conducted the methodological part, but it can be improved by adding a number of samples to include more diversity in their future study. If authors agree, the discussion part can be improvised.

Overall, the article is in good shape with all required data. It can be accepted.

Author Response

Thank you for the positive feedback.  We also greatly appreciate your very helpful suggestions.

We are currently getting more canine tissues, and trying to establish more canine cell lines using conditional reprogramming methods.  Some are from normal tissues, and some are from CPV infected tissues.  Hopefully in a near future we can present our next batch of data in a future publication.